# Tailored Viral-like Particles as Drivers of Medical Breakthroughs

**DOI:** 10.3390/ijms25126699

**Published:** 2024-06-18

**Authors:** Rafael Travassos, Sofia A. Martins, Ana Fernandes, João D. G. Correia, Rita Melo

**Affiliations:** 1Centro de Ciências e Tecnologias Nucleares, Instituto Superior Técnico, Universidade de Lisboa, CTN, Estrada Nacional 10 (km 139.7), 2695-066 Bobadela, Portugal; rafaeljctravassos99@gmail.com (R.T.); sada.martins@campus.fct.unl.pt (S.A.M.); anafsfernandes00@gmail.com (A.F.); 2Departamento de Engenharia e Ciências Nucleares, Instituto Superior Técnico, Universidade de Lisboa, CTN, Estrada Nacional 10 (km 139.7), 2695-066 Bobadela, Portugal

**Keywords:** nanoparticles, virus-like particles, surface functionalization, biomedical applications

## Abstract

Despite the recognized potential of nanoparticles, only a few formulations have progressed to clinical trials, and an even smaller number have been approved by the regulatory authorities and marketed. Virus-like particles (VLPs) have emerged as promising alternatives to conventional nanoparticles due to their safety, biocompatibility, immunogenicity, structural stability, scalability, and versatility. Furthermore, VLPs can be surface-functionalized with small molecules to improve circulation half-life and target specificity. Through the functionalization and coating of VLPs, it is possible to optimize the response properties to a given stimulus, such as heat, pH, an alternating magnetic field, or even enzymes. Surface functionalization can also modulate other properties, such as biocompatibility, stability, and specificity, deeming VLPs as potential vaccine candidates or delivery systems. This review aims to address the different types of surface functionalization of VLPs, highlighting the more recent cutting-edge technologies that have been explored for the design of tailored VLPs, their importance, and their consequent applicability in the medical field.

## 1. Introduction

Nanotechnology can be defined as the design, construction, and characterization of solid colloidal materials within the nanometer range (1–100 nm) [1]. Nanoparticles (NPs) display a high surface-area-to-volume ratio and elevated surface energy, as well as flexible magnetic, mechanical, electrical, and biological properties, rendering them suitable for a plethora of applications [1,2,3]. NPs have been explored as bioremediators [4], sensors, and catalysts in environmental applications [5,6]; as drug delivery tools and imaging agents in medicine [7,8]; as pesticides and fertilizers in agriculture [9,10]; as food sensors; and as electronic devices [11,12]. Among these applications, the biomedical fields are particularly relevant because NPs can potentially overcome the constraints of traditional drug delivery, namely systemic toxicity, inadequate specificity, and poor bioavailability [1,13,14]. Indeed, NPs can increase the stability and solubility of encapsulated substances, enhance the circulation time of the payload in the body, and promote the delivery of desired cargo across cellular membranes [14]. Furthermore, NPs can efficiently carry and deliver therapeutic cargoes, imaging probes, or biological substances to specific targets, paving the way for the development of more effective theranostic strategies [3,15]. However, only a few lipid- and protein-based formulations have been approved for use in clinical settings, despite the promise of NPs in general [15,16]. Each type of nanoparticle has its advantages and challenges, contributing to the expansion of the field of nanotechnology. Among the different types of nanoparticles, virus-like particles (VLPs) emerge as a versatile subclass with enormous potential to revolutionize medicine, diagnosis, and materials science [17,18,19]. Their ability to mimic viruses without causing infection makes them valuable tools for advancing research and applications in various fields.

### 1.1. Virus-Like Particles

VLPs are typically constructed using proteins from a single virus, although structural proteins from diverse viruses may also be employed for VLP assembly [19,20]. These nanostructures were first found in the serum samples of patients with Down’s syndrome, hepatitis, and leukemia, and exhibited antigenic sites at their surface [21]. VLPs range from 20 to 200 nm in size and present a set of advantageous characteristics that make them particularly useful for medical applications [17,18,20,22]. These features include: ease of manufacturing, since they can self-assemble from virus proteins through a spontaneous process [23]; ability to cross cellular membranes through receptor-mediated endocytosis while escaping lysosomal degradation due to their viral nature [24]; immunogenicity derived from their virus-like structure, which deems them an auspicious tool for vaccine development [25]; capacity to improve the stability and solubility of encapsulated cargo as it protects against degradation [26]; biocompatibility due to their biological nature [27]; and ease of modification through chemical [28] or genetic processes [29].

VLPs can be assembled with proteins and a wide variety of viruses such as human papillomavirus (HPV) [30], human immunodeficiency virus (HIV) [31], the Norwalk virus [32], and the influenza virus [33]. Moreover, VLPs can be constructed in different expression systems, namely prokaryotic (bacteria) [34], eukaryotic (plant cells [35], insect cells [36], yeast [37], and mammalian cells [38]), and cell-free platforms [39].

To improve the efficiency, potency, and cargo-loading capacity, current studies have mainly focused on understanding the self-assembly mechanisms of VLPs [40,41]. This process may be aided by different types of molecules, such as nucleic acids and supporting proteins [42].

After assembly, the surface of VLPs can be functionalized with a myriad of moieties, including ligands for T-cell receptors [43], antibody fragments [44], polysaccharides [45], polymers [46], enzymes [47], and natural amino acids [48], among others.

Although VLPs are devoid of viral genetic material, they can maintain certain features of the original virus, namely cellular receptor affinity, host cell entry points, and immunogenicity [49]. Consequently, VLPs can prompt cellular and humoral responses. Furthermore, it is possible to heighten the immune response elicited by VLPs through chemical or genetic modification of the surface of the VLP with different antigens (such as proteins or short peptides) [50]. The ability to incorporate multiple molecules onto the surface of VLPs makes them multivalent nanoplatforms that can function as vaccine formulations, imaging probes, and/or therapeutic tools.

### 1.2. VLP Structure

There are three main classes of VLPs: enveloped, non-enveloped, and chimeric [19]. Enveloped VLPs are structurally complex, attaining a lipidic membrane from the host platform during assembly and budding and, consequently, requiring expression in eukaryotic platforms [19]. Examples of enveloped VLPs include HIV VLPs, influenza VLPs, or Ebola VLPs [19]. Non-enveloped VLPs can harbor single or multiple capsid proteins and be single-, double-, or triple-layered, and can thus be produced in both prokaryotic and eukaryotic systems [19]. HPV VLPs and Norwalk VLPs are examples of non-enveloped VLPs [19]. The classification and characteristics of some enveloped and non-enveloped VLPs are depicted in Table 1.

Lastly, chimeric VLPs harness structural proteins from different viruses for assembly [19]. These VLPs are mostly explored for vaccination purposes, and due to their chimeric nature, are not described with surface functionalization. Chimeric VLPs include VLPs combining HIV-1 Gag and a fusion rabies glycoprotein [59] and VLPs with a Hepatitis B core bearing a SARS-CoV-2 epitope [60], among others).

The assembly of VLPs is affected by their structure and composition. Furthermore, some factors can compromise VLP assembly, as shown in Figure 1 [42].

The presence or absence of certain natural amino acids, as well as the structure acquired by the peptide chain, is crucial for VLP assembly [62]. Mutagenesis of structural proteins can lead to incomplete assembly or hinder the formation of VLPs [62]. Interestingly, one study has reported that low temperatures are favorable for B19-VP2 VLP self-assembly since they can diminish the occurrence of undesired processes, such as aggregation and protein degradation [63]. On the contrary, temperatures in the physiological range also contribute to VLP assembly [64], indicating that temperature is pivotal for VLP construction. The pH interferes with capsid-protein charge, and most studies indicate that VLP assembly is optimal at physiological pH [63,65,66]. Nonetheless, adequate VLP formation has been described for both acidic [23] and alkaline [67] pH conditions. Salts also affect protein stability through interactions with charges on the capsid protein surface, making ionic strength a key factor in VLP assembly [42]. Finally, cargo encapsulation is highly linked with VLP assembly, and cargo change and structure need to be considered when developing VLPs [42]. All these factors, together with surface functionalization, must be managed to attain functional VLPs.

### 1.3. VLP Downstream Processing

In order to guarantee appropriate safety and effectiveness for clinical usage, downstream processing for VLP purification is an essential step [68]. The initial step of the purification process following cell harvesting is determined by the VLPs’ capacity to be discharged into the extracellular media. Cell lysis or another extraction technique could be necessary to disrupt the cells if the VLP is not released successfully, despite the fact that in certain documented cases—such as influenza VLPs generated in insect cell culture—the particles are released into the media without the need for particular precautions [69]. Generally, the strategy used is to create a cloned gene that expresses a protein with an efficient signal peptide that the secretory system will recognize and thus assist in VLP release [68,70].

A clarity phase is carried out to eliminate cell debris and aggregates from VLP preparations to streamline the procedures and lower the expenses of the purification process [68]. One crucial step that dramatically lowers the bulk volume and increases the ratio of VLP concentration in relation to other cellular contaminants is the capture and concentration of VLPs [68]. Techniques such as depth filtration, microfiltration, tangential flow filtration (TFF), cell sedimentation, UF/DF (Ultrafiltration/Diafiltration), and TFF with membranes or hollow fibers are often used for separating VLPs from host cell contaminants such as cell debris, digested DNA, or medium components [68]. VLPs may also be captured using the bind-and-elute method using affinity chromatography, ion-exchange chromatography (IEC), and hydrophobic interaction chromatography (HIC) [71,72].

To lower DNA and endotoxin levels, further intermediate purification processes including IEC, HIC, and ultra-centrifugation are frequently needed [68].

An optional procedure used to improve the stability, homogeneity, and immunogenicity of the VLP product is disassembly and reassembly. For this phase, titration or cross-flow filtration are the selected methods [68]. All impurities left behind from processing must be eliminated in the polishing phase, which is the last stage of purification [68]. IEC, size exclusion chromatography (SEC), and UF/DF (often the crossflow technique) are frequently used for this [68,72]. The preparations are sterilized before being finalized by filtering through sterile-grade filters with a suitable pore size [72].

A downstream processing approach for VLP purification is summarized in Figure 2.

An updated list of VLP surface functionalization is provided in the following sections, together with information on their potential prophylactic, diagnostic, and therapeutic applications.

## 2. Types of Surface Functionalization

Owing to their structural characteristics, VLPs can be designed by expressing proteins, peptides, nucleic acids, imaging probes, drugs, and other NPs [74,75,76] on their surface. They may also have target ligands that alter the inherent tropism of VLPs by making them specific to a particular cell, tissue, or organ [44,77]. Both chemical and genetic approaches can be employed for the surface modification of VLPs [28,29] (Figure 3).

Functionalization may differ according to the nature of the VLP. The functionalization of non-enveloped VLPs may be achieved through covalent conjugation to amino acid residues on the surface [78]. Cysteine, glutamic acid, and lysine are natural amino acids that have been explored for functionalization [78]. For example, one study reported the conjugation of fluorescent probes to cysteine residues present on tobacco mosaic VLPs for the development of photovoltaic devices [79]. Destito et al. harnessed surface lysine residues on cowpea mosaic VLPs for the conjugation of folic acid to achieve efficient targeting of cancer cells [80]. However, the location of the amino acid residues within the VLP must be considered for reactivity or surface accessibility [81]. The introduction or elimination of amino acids at crucial locations using recombinant DNA technologies can be used to overcome such limitations [81]. Indeed, Thong et al. introduced a reactive cysteine on the inner surface of MS2 VLPs for the incorporation of a fluorescent dye [81]. Interestingly, Patel et al. employed the amino acid mutation method for the construction of MS2 and Qβ VLPs with surface-exposed methionine analogs comprising azide and alkyne side chains for the subsequent conjugation of azide- and alkyne-harboring molecules, namely an antibody fragment and the granulocyte-macrophage colony stimulation factor [44]. Alternatively, fusion between virus capsid proteins and the desired proteins and peptides can be performed, enabling the internal or external framing of non-enveloped VLPs [77]. This technique allows the appropriate targeting moiety to be directly attached to viral capsid proteins, despite being circumscribed to peptides and proteins [78]. The size of these peptides must also be reduced to prevent interference with VLP function [78]. For example, epidermal growth factor was genetically fused to the C terminus of the Qβ capsid protein to form epidermal growth factor receptor (EGFR)-specific VLPs [77], and only a relatively small fraction of incorporated fusion proteins provided effective cell binding and internalization, leaving most of the outer surface of the nanoparticle for subsequent chemical derivatization. Another study focused on HPV VLPs and depicted the fusion of green fluorescent protein (GFP) to the N or C terminus of L1 and L2 capsid proteins, to attain VLPs that can function as delivery vehicles of foreign proteins [82]. Recently, Guo et al. used genetic recombination to insert a coronavirus epitope into the Qβ capsid protein for the development of a chimeric VLP vaccine [83]. Lastly, it should be noted that non-enveloped VLPs can undergo non-covalent alterations by electrostatic [84,85], protein–protein [86,87], protein–nucleotide [75,88], and protein–metal [89,90] interactions. Because non-covalent modifications are frequently reversible, this might be problematic when prolonged stability is warranted.

Given the lipidic nature of their envelope, enveloped VLPs are better adapted for the display of membrane proteins [91]. This type of VLP can also undergo pharmacological or genetic modifications, with the latter taking place when a desired protein is fused to a full-length viral protein or a transmembrane domain [92,93]. This can be achieved through loop insertion [94], N- or C-terminal modifications [94], or amino acid residue mutations [44]. For example, Boxus et al. reported the construction of several HPV chimeric VLPs through the insertion of L2 epitopes into the capsid surface DE loop of HPV-16 or HPV-18 L1 proteins for the development of a novel vaccine formulation against HPV [95]. Regarding N- or C-terminus modification, Gordon et al. described the fusion of the carboxy terminus of the circumsporozoite antigen derived from *Plasmodium falciparum* to the N-terminus of the hepatitis B surface antigen, generating chimeric VLPs that were later commercialized as the Mosquirix^®^ vaccine against malaria [94,96].

A protein can undergo biochemical reactions after translation, known as post-translational modifications (PTMs), leading to changes in its properties [97,98,99]. PTMs are obtained through the enzymatic binding of groups such as acetyl, glycosyl, phosphoryl, and methyl, which are covalently added to a single type of amino acid or multiple amino acids during the process [97,98,99,100]. These groups can lead to changes in protein activity and stability, along with changes to the physicochemical and conformational properties [97,98,99]. Consequently, PTM dysregulation leaves its mark on a wide range of illnesses. Phosphorylation is a key PTM link to cardiovascular disease, Alzheimer’s disease, and diabetes. The Tau protein in Alzheimer’s disease is phosphorylated at forty sites, correlating with disease phases [101]. Methylation, acetylation, glycosylation, and S-nitrosylation are also linked to neurological and cardiovascular conditions [101,102], while diabetes is associated with dysregulated glycosylation [103]. Ubiquitination is crucial in rare congenital illnesses such as von Hippel–Lindau syndromes [101]. Key PTMs in diseases include ubiquitylation, SUMOylation, prenylation, S-palmitoylation, and glycosylation, which also enhance peptide pharmacological properties and folding [97,98,99]. Several studies have explored the role of PTMs in VLPs [104,105]. Balieu et al. focused on the N-glycosylation pattern of the SPIKE protein within SARS-CoV-2 VLPs developed in Nicotiana benthamiana plants, and the results revealed that the overall VLP structure exhibited complex N-glycans that are similar to those expressed in mammalian platforms [104]. This highlights the potential of plants to produce novel vaccines. Another study reported that SUMO modifications are relevant for the proper assembly of foot-and-mouth-disease (FMDV) VLPs, as they prevent inadequate aggregation and increase the solubility of capsid proteins [105].

## 3. Biomedical Applications of Functionalized VLPs

The surface functionalization of VLPs renders them flexible platforms that can be explored in different biomedical fields, such as drug delivery, vaccination, and imaging, as well as other applications, which will be presented and discussed in subsequent sections.

### 3.1. Drug Delivery

The precise and selected delivery of drugs to target tissues, notably anticancer drugs, is a tremendous challenge faced by medicinal chemists. One of the most auspicious strategies to surmount such limitations is to encapsulate drug molecules inside a nanoparticle that specifically targets cancer cells, which may also improve the stability of the drug in circulation [106]. Surface-functionalized VLPs with active compounds may present several advantages, namely improved pharmacokinetics, optimal biodistribution, and reduced adverse effects [19,107]. Most studies that have explored VLPs as drug delivery systems have focused on microRNAs as cargo [75,108,109,110,111,112], but the VLP-mediated delivery of other agents has also been described [23,75,113,114].

Pan et al. have constructed a microRNA delivery platform grounded on bacteriophage MS2 VLPs [108]. This study aimed to investigate the potential of these nanoplatforms as delivery systems. The VLPs were constructed by the fusion of pre-microRNA-146a (miRNA-146a) to an MS2 cistron, which then interacted with the bacteriophage coat protein for assembly [108]. Subsequently, VLPs were conjugated to a peptide derived from the HIV trans-activator of transcription (TAT), which has been described as an effective cell-penetrating peptide [108]. VLP concentrations in the nanomolar range were assessed both in vitro and in vivo [108]. The results showed that the VLPs successfully transferred pre-miR146a to different cells and tissues, increasing the expression of miRNA-146a both in vitro and in vivo, and subsequently suppressing its target gene [108]. This group has also focused on the effects of miR-146a on osteoclastogenesis and systemic lupus erythematosus (SLE) [109,110]. In the first study, MS2 VLPs were employed as delivery systems for miR-146a in human peripheral blood mononuclear cells (PBMCs) [109]. The addition of MS2-miR-146a-VLPs to PBMCs prompted the upregulation of miR-146a, highlighting the potential of MS2 VLPs as delivery platforms [109]. Furthermore, upon treatment with osteoclastogenesis-inducing cytokines, osteoclastogenesis-specific genes were downregulated in PBMCs transfected with MS2-miR-146a-VLPs, together with a reduced number of osteoclasts [109]. In the study, the authors investigated the role of miR-146a in systemic lupus erythematosus (SLE) and treated lupus-prone mice with MS2-miR146a VLPs [110]. Treatment of mice with TAT-conjugated MS2-miR146a VLPs led to increased levels of mature miR146a, which translated into a significant reduction of both auto-antibody expression and proinflammatory cytokine levels [110].

Another study reported a microRNA delivery platform based on MS2 VLPs functionalized with a GE11 polypeptide [111]. This polypeptide binds to the epidermal growth factor receptor (EGFR), which is overexpressed in several malignancies, namely hepatocellular carcinoma (HCC) [115,116]. Moreover, the maternally expressed gene 3 (*MEG3*), which encodes a long non-coding RNA, has shown potential as a tumor suppressor in several carcinomas, including HCC [117]. The scarcity of delivery vehicles for this long non-coding RNA spurred the study of VLPs as a valid option [111]. To package the MEG3 complementary DNA (cDNA) sequence, two mutated pac site sequences, which are viral sequences responsible for nucleic acid packaging, were inserted into MEG3 cDNA [111]. Then, the Sulfo-SMCC crosslinker was used to form a bridge between the GE11 polypeptide and MS2 VLP [111]. This study demonstrated that this is a promising delivery system in cancer therapy, as it was possible to achieve swift, effective, and safe delivery of MEG3 to EGFR-positive HCC cell lines, with no activation of downstream EGFR pathways [111].

Wang et al. carried out new studies to avoid the use of a crosslinker, as it is expensive and requires preparation under strict conditions, and to increase the efficiency of the delivery system [112]. The authors took advantage of MS2 VLPs as delivery systems for microRNA-122 (miR-122) and placed the TAT sequence on the surface of the VLPs by phage surface display instead of using chemical crosslinkers [112]. It has been observed that miR-122 is deregulated in HCC and negatively regulates the expression of genes that partake in cancer cell proliferation, migration, and invasion [118]. The authors therefore incorporated the TAT peptide into the N-terminus of the MS2 coat protein dimer and added a pac site to the pre-miR122 for packaging [112]. The results showed that these recombinant MS2 VLPs containing TAT (MS2-TAT-miR122 VLPs) are efficient, penetrating the cytomembrane and inhibiting HCC through the delivery of miR-122. This shows that this type of platform can be explored for drug delivery and as a promising treatment for HCC [112].

Interestingly, Ashley et al. reported the construction of MS2 VLPs for the selective delivery of different types of moieties, namely chemotherapeutic agents, small interfering RNA (siRNA), and protein toxins to HCC [75]. To achieve selectivity, the VLPs were functionalized with an HCC-targeting peptide, SP94, through a crosslinker with a polyethylene glycol (PEG) spacer [75]. Encapsulation was conducted through the conjugation of a pac site to the different cargoes [75]. VLPs were thus loaded with quantum dots, three chemotherapeutic agents (doxorubicin (DOX), cisplatin, and 5-fluorouracil (5-FU)), and a ricin toxin A-chain (RTA and a siRNA that negatively regulates the expression of cyclin family members) [75]. The results were as follows: VLPs that contained DOX, cisplatin, and 5-FU were able to selectively kill HCC cells with drug concentrations under 1 nM; VLPs comprising a siRNA cocktail were able to prompt growth arrest and apoptosis of HCC cells with concentrations under 150 nM; VLPs loaded with RTA and co-functionalized with SP94 and a peptide that spurs endosomal escape were capable of killing all cells with concentrations as low as 100 fM [75]. These studies highlight the flexibility of MS2 VLPs as delivery vehicles.

Another study described the attachment of DOX to VP6, a rotavirus capsid protein, for VLP assembly [23]. Lactobionic acid (LA), a ligand of asialoglycoprotein receptors (ASGPRs), is overexpressed in the membrane of HCC cells [119]. In vitro experiments demonstrated that these VLPs (DVLPLAs) were internalized by a hepatoma cell line, and most cells were killed with a concentration of 10 μg mL^−1^ of DOX when loaded on VLPs [23].

Kato et al. explored *Rous sarcoma* VLPs (RSV VLPs) as drug delivery platforms through functionalization with an hCC49 single-chain fragment variable (scFv), which specifically targets the tumor-associated glycoprotein-72 (TAG-72) on the membrane of colon carcinoma cells [113]. The cytoplasmic and transmembrane domains of influenza A hemagglutinin were used to fuse the scFv to the VLPs, which were expressed in silkworm larvae [113]. Electroporation was used to encapsulate DOX, and specific targeting was assessed [113]. These results indicate that hCC49 scFv-presenting RSV VLPs can target colon carcinoma cells and efficiently deliver DOX [113].

More recently, a novel technology called SpyCatcher/SpyTag has been explored to develop targeted HBV VLPs [114]. The SpyCatcher/SpyTag system comprises a genetically encoded protein, SpyCatcher, which is designed to spontaneously and covalently bind to its peptide partner SpyTag [120].

As such, VLP proteins genetically fused to SpyCatcher can be easily functionalized to moieties linked to SpyTag [120]. Hartzell et al. inserted the SpyCatcher into the c/e1 loop of HBV and tested different moieties with the SpyTag for VLP decoration [114]. For example, the authors selected the GE11 peptide, which targets EGFR, to decorate the HBV VLPs and thus specifically target inflammatory breast cancer cells [114]. Furthermore, a dimeric prodrug-activating enzyme was loaded into the VLPs, which enabled the specific uptake of the VLPs by the breast cancer cells and subsequent enzyme delivery and cell killing [114].

### 3.2. Vaccines

VLPs are suitable platforms for vaccination, deeming it their most common medical application [121,122]. VLP-based vaccines can provide more advantages than traditional vaccines because they structurally mimic viruses, elicit a memory response, and have several epitopes in their structure, which prompt humoral and cell-mediated responses, enhancing immunity [121,122]. Furthermore, since VLPs do not have a viral genome, they cannot infect or replicate, thus not compromising safety [121,122]. Several VLP-based vaccines against viruses have been developed in recent years and are now commercially available (Table 2).

VLPs can function as stand-alone vaccines against their native viruses, relying on their virus-like structure to elicit an immune response [121,122]. Notwithstanding, VLP functionalization has also been described for the development of vaccination formulations, namely against cancer. An example of the application of surface-functionalized VLPs in vaccines is the previously discussed VLPs developed by Patel et al. [44], which are derived from bacteria-infecting viruses (bacteriophages MS2 and Qβ). This bestows them with an important competitive edge over VLPs derived from human-infecting viruses, such as HIV, because they are less susceptible to pre-existing antibodies in the human population, potentially preserving the immunogenicity of these vaccines. These VLPs were produced with methionine analogs that contain azide (azidohomoalanine: AHA) and alkyne (homopropargylglycine: HPG) terminal groups, which are non-natural amino acids (nnAAs) and are exposed at the surface for conjugation, which occurred in a single step catalyzed by Cu(I) using click chemistry. Consequently, VLP functionalization with three different ligands was achieved in one step. This study depicted VLP production through direct conjugation with antigens from idiotype antibody fragments, cytokine GM-CSF, and immunostimulatory CpG DNA to produce new candidate tumor idiotype vaccines for the treatment of B-cell lymphoma [44].

VLPs have also been modified with the human epidermal growth factor receptor 2 (HER2). Palladini et al. harnessed the SpyCatcher/SpyTag system to construct multivalent HER2 extracellular domain (ECD)-presenting AP205 VLPs [135]. The immunogenic and therapeutic potential of this vaccine formulation was assessed in mice bearing HER2-overexpressing mammary carcinoma cells and compared with a traditional DNA vaccine [135]. This vaccine was able to elicit strong anti-HER2 auto-antibody responses, limit the spontaneous development of HER2-overexpressing carcinomas, and hamper the growth of HER2-overexpressing tumors introduced into wild-type mice [135]. The VLP vaccine showed better results than the DNA vaccine, and no significant side effects were observed in the treated mice, which corroborates the potential of VLPs as cancer vaccines [135].

Salazar-González et al. designed B19 human parvovirus VLPs with chimeric VP2 proteins and functionalized them with two insulin-like growth factor-1 receptor (IGF-1R) epitopes through fusion with the N-terminus of VP2 [136]. IGF-1R has been related to angiogenesis in different malignancies, and epitopes P8, a B-cell epitope within the extracellular domain of the receptor, and 249, a T-cell receptor that binds to both major histocompatibility complexes, have been identified as relevant epitopes, and were thus selected for the development of VLP vaccines [136]. Several chimeric multi-epitope VLPs were thus generated and were employed in a prophylactic setting of four weekly immunizations preceding 4T1 cell inoculation in mice [136]. Then, four weekly VLPs were administered via intraperitoneal injections, and this regime was able to protect mice from tumor formation and growth following cell inoculation [136]. It was possible to observe specific antibodies against the presented epitopes; however, there were no significant T-cell responses, indicating that further studies are required [136].

The recent COVID-19 pandemic has also spurred the development of novel vaccination strategies, and one study described the fusion of the SARS-CoV-2 S protein to the Gag protein of either HIV-1 or simian immunodeficiency virus (SIV) in a mRNA vaccine for the assembly of chimeric VLPs [137]. This vaccine was inoculated into mice and its effects in an in vivo model were assessed [137]. A high level of S-trimer-binding and neutralizing antibodies was observed, demonstrating that this platform can be explored as a vaccine against different pathologic agents [137].

### 3.3. Imaging

Imaging is pivotal for diagnosis and visualization of disease location and progression, providing greater monitoring of the disease in the patient and thus leading to more successful treatment [138,139]. Given the high biocompatibility and easy surface functionalization of VLPs, their combination with several imaging agents has been described [18,138].

Li et al. selectively modified the surface-coating protein P8 of the bacteriophage M13 on the lysine, aspartic, or glutamic acid side chains, and used tyrosine residues for conjugation [140]. After surface modification, the bacteriophage M13 undergoes a double modification with fluorescein dyes (reactive fluorescein N-hydroxy succinimidyl ester: FL-NHS) and folic acid (FA) [140]. A first incubation with FL-NHS was carried out to give rise to a VLP dye/phage (M13-FL), followed by dialysis. Then, M13-FL was functionalized with folate-azide using a tyrosine conjugation method [140]. The results obtained by Li et al. show that M13 was double-modified (FL-M13-FA) and through three-dimensional imaging, it was possible to observe that the FL-M13-FA VLPs were in the cytoplasm of KB cells and could be explored as fluorescent probes for imaging in this type of human cell [140].

Another example is the work conducted by Sun et al., which focused on the development of simian virus 40 (SV40)-based VLPs carrying near-infrared quantum dots for imaging and peptides for targeting atherosclerotic plaques [141]. Atherosclerosis is a systemic pathology that affects arteries and is characterized by the deposition of plaques in arterial walls, which may spawn myocardial infarction and stroke [142]. Traditionally, the diagnosis of this disease was only carried out in later stages, but novel imaging modalities, namely optical imaging, are currently under study [143]. Within optical imaging, near-infrared quantum dots emerge as promising candidates due to their detection sensitivity in deep tissues [144]. Sun and colleagues genetically inserted an atherosclerosis-targeting peptide into the HI loop region of the SV40 major capsid protein VP1, and a thrombin inhibitor peptide (Hirulog) into the N-terminus of VP1 [141]. The authors also developed VLPs harboring both a peptide that targets fibrin at an advanced stage of atherosclerosis and Hirulog, and VLPs that only contained a peptide that targets vascular cell adhesion molecule-1 (VCAM-1), a marker for the original state of atherosclerosis [141]. These engineered proteins underwent self-assembly together with near-infrared quantum dots to form multifunctional VLPs, which were then evaluated in an in vivo model [141]. The multifunctional VLPs were intravenously injected into mice, and the quantum dots were able to elicit sufficient signals for imaging of atherosclerotic plaques in mice, and the targeting moieties allowed for specificity to the aorta, namely in sites where atherosclerotic plaques are typically formed [141]. Immunohistochemistry analysis revealed that the targeted nanoplatforms co-localized with their targets, and Hirulog was able to exert its function and treat thrombin [141]. All these data underline the functionality and practicality of VLPs as theranostic platforms [141].

A VLP for PET imaging was designed upon conjugation of an antibody against EGFR to MS2 capsid proteins and consequently labeled with copper-64 (^64^Cu) [145]. To construct this tailored nanoplatform, an oxidative coupling strategy of aminophenols and anilines was employed [146]. Anilines were incorporated into MS2 viral capsids via the introduction of *p*-aminophenylalanine on the outer surface, and aminophenol was inserted into the antibody through non-site-specific lysine modification with nitrophenol-NHS-ester [146]. The two were then coupled through an oxidative reaction [146]. A maleimide chelator was then added to the VLPs for radiolabeling with ^64^Cu, and the resulting VLPs were evaluated both in vitro and in vivo [145]. In vitro studies revealed specific binding to the EGFR receptor present on the membrane of different cancer cell lines [145]. The radiolabeled EGFR-targeting VLPs were injected into mice bearing tumor xenografts and ex vivo PET imaging was performed to assess VLP location [145]. Moderate tumor uptake and long circulation periods were observed, but no significant specific uptake was observed, indicating that further improvements are required to achieve a targeting VLP that is useful for PET imaging [145].

More recently, physalis mottle virus (PhMV) VLPs were loaded with the fluorescent dye Cy5.5 and gadolinium (Gd)(III) complexes for near-infrared fluorescence and MRI, respectively, and functionalized with a targeting peptide [147]. Thiol-maleimide chemistry was used to conjugate the DOTA chelator (and subsequent radiolabeling with Gd(III)) and Cy5.5 to the inner surface of the VLPs. In addition, amine-*N*-hydroxysuccinimide ester chemistry was used to conjugate the DGEA peptide, which targets integrin α_2_β_1_, a receptor that is overexpressed in the prostate cancer cell line PC-3 [147]. PC-3 cells were treated with modified VLPs, and it was possible to observe a high fluorescent signal and subsequent high VLP cell uptake [147]. VLPs were injected into mice harboring PC-3 tumors, and imaging analysis revealed that the VLPs were efficiently taken up by the tumors [147].

### 3.4. Biosensing

Biosensing is based on receptor-transducer devices capable of transforming a biological signal into an electrical response and consequently originating a measurable signal [148]. These instruments typically comprise an analyte, a bioreceptor, a transducer, electronics, and a display unit, and are utilized to measure pH levels, pathogen concentrations, and toxic chemicals, among others [148]. VLPs, besides vaccination, imaging, and drug delivery, have also recently been explored for biosensing.

Mieloch et al. developed biomimetic VLPs with superparamagnetic iron oxide nanoparticles and the receptor binding domain (RBD) of SARS-CoV-2 [149]. The bioactivity of the particles was assessed, and it was possible to observe that the VLPs were capable of binding to SARS-CoV-2-specific targets, such as angiotensin-converting enzyme 2 (ACE2) and toll-like receptor 4 (TLR4) [149]. Furthermore, these particles were capable of capturing SARS-CoV-2-targeting antibodies with high specificity and are therefore presented as an alternative to typical COVID-19 detection kits [149]. However, these data are preliminary, and further tests on the ability to immobilize these particles on detection structures are still required [149].

Among the different biosensors under development for disease diagnosis, nucleic acid-based fluorescent biosensors have gained relevance, in particular for living sample analysis [150]. Within these biosensors, the hybridization chain reaction (HCR), an enzyme-free isothermal amplification method, is highlighted due to its potential in living imaging [151]. The main principle of HCR is based on the assembly of stable DNA monomers following exposure to target DNA sequences [152]. Basically, two stable DNA hairpins are placed in solution, and the introduction of initiator sequences spurs hybridization that results in double strands, whose molecular weight is inversely proportional to the initiator concentration [152]. HCR can be conjugated to aptamer triggers and therefore function as an amplifying transducer in biosensors [152]. Traditional HCR, however, is not highly sensitive due to its single linear extension circuit and cannot be easily taken up by biological systems due to the negative charge of nucleic acids. Chen et al. attempted to circumvent these issues by designing a novel strategy that harnesses VLPs [153]. First, a stack design was employed to develop a hierarchical HCR (H-HCR) that can graft multiunit HCR products [153]. The elements of this H-HCR were then packaged into VP3 (structural protein of the human bocavirus 1 capsid) VLPs engineered to present arginine–glycine–aspartic acid (RGD) peptides, which target α_v_β_3_ integrin (a cancer biomarker) [153]. Afterwards, a multibranched DNA scaffold was designed to bear the H-HCR components and heighten the H-HCR produced signals [153]. Additionally, a photocleavage linker (PC-linker) was introduced to bestow spatiotemporal light-powered behavior, and thus ensure that imaging signals are not lost during delivery [153]. The authors designated this method as a genetically engineered VLP-armored and multibranched DNA-scaffold-corbelled H-HCR and tested it for the detection of a cancer-associated mRNA marker (miRNA-155) in vitro, as well as for imaging in living cells [153]. This method proved to be highly specific and ultrasensitive and is capable of providing strong imaging in both healthy and cancerous cells, and well as in tumor-bearing mice [153]. This innovative strategy introduces novel paths to achieve stronger and more robust disease diagnosis.

Another study reported the design of a coulometric enzyme immunoassay for the detection of immunoglobulin M (IgM) against Hantaan viruses (HTNVs) by exploring VLPs as detecting moieties [154]. First, the authors functionalized screen-printed carbon electrodes (SPCEs), which are widely used electrodes in point-of-care biosensors, with paste-exfoliated graphene (a carbon material presenting good electrical conductivity) bearing a carboxylic acid group and a thionine mediator [154]. Then, an antibody against HTNV IgM was immobilized onto the SPCE surface [154]. HTNV VLPs were employed as antigens for the second recognition of anti-HTNV IgM antibodies, and these VLPs were functionalized with horseradish peroxidase (HRP) for coulometric measurement of hydrogen peroxide reduction mediated by thionine following the application of an electrical potential to the electrode system. This potential is necessary to drive the electrochemical reduction reaction of hydrogen peroxide catalyzed by HRP [154]. This system could detect anti-HTNV IgM in human serum with high selectivity and sensitivity, and highlights the potential of VLPs in biosensing, especially in disease marker detection [154].

## 4. Challenges of VLP-Based Approaches

Despite the promising results exhibited by VLPs, it is important to highlight that only a few VLP-based formulations have been approved and commercialized. These formulations were exclusively approved for vaccination, which is linked to the high immunogenicity of VLPs. This can severely hamper their use as imaging and/or drug-delivery tools. Furthermore, there are several hurdles related to VLP and expression host selection, functionalization, scalability, downstream processing, and stability. All these factors must be taken into consideration when developing novel VLP-based tools.

### 4.1. Selection of VLP Type and Expression Platform

Most of the marketed VLPs are non-enveloped due to their simplicity. Production of non-enveloped single-protein capsid VLPs is typically performed in *Escherichia coli* or yeast, and both systems allow for high production yields and easy scale-up [155]. Furthermore, yeast expression systems can provide PTMs, which can play a pivotal role in VLP function [155]. Interestingly, due to their simplicity, these VLPs can be produced in a cell-free environment. Non-enveloped multiple-capsid protein VLPs, on the other hand, are typically produced in the baculovirus-insect expression system [155]. This system can also provide high yields and PTMs, but it is not as easily scalable and regulated as *E.coli* and yeast and requires more complex purification processes [155]. Production in microbial cells typically entails cell lysis, which can release cell-derived impurities and consequently affect downstream processing [156]. Moreover, the production process of non-enveloped VLPs typically generates non-uniform particles [157]. To overcome these hurdles, these VLPs typically undergo in vitro disassembly and reassembly, which has proved to improve stability [158] and immunogenicity [159]. The main advantages of non-enveloped VLPs include their well-characterized nature and their stability, given that they are less sensitive to external conditions (purification, temperature, or sheer force) than enveloped VLPs [160]. This explains why most FDA-approved VLPs are non-enveloped. Nonetheless, they are not as suitable for the presentation of membrane proteins as enveloped VLPs due to the origin of their envelope [91]. Even so, the advantages presented by these VLPs explain why most VLP-based platforms developed in recent years for drug delivery and imaging are non-enveloped.

Enveloped VLPs are structurally more complex than non-enveloped VLPs since they attain their lipidic envelope from their expression platform during assembly and budding [19]. Indeed, this type of VLP can present one or several glycoproteins on their lipidic membrane, which function as target antigens and can therefore elicit an immune response and prompt the production of neutralizing antibodies [19,161]. For this reason, the production of enveloped VLPs requires eukaryotic expression platforms [19]. The major issue with enveloped VLPs is their poor biophysical characterization, given that they are not as structurally uniform as non-enveloped VLPs [155,160]. Moreover, as mentioned above, enveloped VLPs are more sensitive to external conditions than non-enveloped VLPs, which means that their immunogenicity, integrity, and stability can be more easily compromised [160]. Enveloped VLPs also require correct folding and glycosylation to adequately function as vaccine formulations [160]. These challenges deem enveloped VLPs more difficult to work with and consequently implement in a clinical setting; however, strategies such as computational design, genetic engineering, in vitro assembly, and novel processing and purification methods can be explored to circumvent these limitations [160].

Regarding expression hosts, each platform has its advantages and disadvantages, which are depicted in Table 3.

### 4.2. VLP Preparation

VLP production typically entails three steps: production (or upstream processing), purification (or downstream processing), and formulation (for VLP-based vaccines) [19,20]. Production starts with the cloning of the viral genes that encode the proteins of interest, followed by expression and assembly in the expression systems [19,20]. The produced VLPs are then transferred to the solution through cell lysis [19,20]. Clarification is performed to eliminate most aggregates and cell contaminants [19,20]. Further purification is typically required to attain integral VLPs, followed by a concentration step, and a final polishing step is performed to remove any remaining residual nucleic acids and proteins [19,20]. Formulation typically entails the incorporation of adjuvants and extra ingredients to obtain a safe and efficient vaccine [19,20].

Besides the selection of the most adequate expression system, it is also critical to select the most suitable culture mode since it affects concentration, productivity, and yield [54,160]. There are three different culture modes for VLP production: continuous cultivation, batch, and fed-batch [54,160]. Continuous cultivation typically ensures the highest productivity since it relies on cultures in a log phase, and therefore allows continuous production [54,160]. This culture mode has been reported for VLP production in different expression platforms, such as insect cells [162] and mammalian cells [163]. The main advantages of this culture mode include a short turn-around period, a reduced number of production steps, and continuous downstream processing [54,160]. Within continuous production, there are different methods: chemostat, morbidostat, stressostat, and turbidostat [54,160]. Among these, the chemostat method is the most frequently used mode as it permits the modulation of growth rates and product formation. Notwithstanding, this method is also associated with a high contamination risk and low product concentrations [54,160]. In the batch mode, all the required nutrients are added only at the start of the culture process, and this brings a low contamination risk [54,160]. The main drawbacks of such a method include low viral titers, low productivity, temperature sensitivity, and product and substrate inhibition [54,160]. Substrate inhibition can be overcome by diminishing the lag phase, which is what occurs in the fed-batch mode [54,160]. In this method, the exponential growth phase can be augmented by the selective incorporation of nutrients, and it is possible to increase the period of culture to attain higher cell densities [54,160]. Nonetheless, scale-up is costly since there is an inefficient use of culture medium [54,160]. Factors such as cost, cell growth, yield, contamination, particle size, and cell lysis, among others, need to be considered when selecting the culture mode [54,160]. It is also important to note that these culture modes were only tested to produce VLP-based vaccines. The studies that report the design and evaluation of VLPs for other applications, namely imaging and drug delivery, are carried out on a small scale and do not account for scale-up. Studies focused on the production process of VLPs for such applications are thus required to determine whether it is possible to attain such VLPs with high stability and productivity for further implementation in the medical field.

Regarding VLP assembly, different factors are at play. Single- or multi-capsid proteins must assemble in expression platforms or cell-free systems, and this assembly may rely on interactions with scaffolding proteins and/or nucleic acids [42,160]. For example, it has been reported that adeno-associated virus (AAV) relies on the assembly-activating protein (AAP) for VLP assembly [67]. Another example is HIV-1 Gag VLPs, which utilize RNA as a scaffold for assembly [164]. Such molecules must be carefully considered during VLP assembly because they can severely affect VLP applications by increasing their immunogenicity [42]. Moreover, factors such as ionic strength, pH, temperature, and stoichiometry heavily influence VLP assembly, so optimal conditions must be fine-tuned to avoid VLP malformation [42,160].

Downstream processing is a critical step in VLP preparation as it allows the separation of a myriad of proteins from the desired VLPs [42,160]. When VLPs are obtained through transduction, viral vectors are used to deliver genetic material to cells, which produces proteins of similar sizes and molecular weights, consequently making the separation process more burdensome [160]. Transfection, on the other hand, does not rely on viral vectors, facilitating separation [160]. Serum supplements possess a high protein content, which can hamper purification, and so the utilization of serum-free media may also be beneficial [160]. Regarding cell lysis, as previously mentioned, this step is only necessary for non-secreting expression systems, as certain insect and mammalian cells secrete VLPs into the cell supernatants [20]. For non-secreting platforms, treatment with detergent-bearing solutions is sufficient for most eukaryotic cells but other platforms, such as bacterial, yeast, and plant cells, possess a cell wall and therefore require mechanical treatment for VLP release, which could hamper their integrity [20]. Ultra-centrifugation is typically used to achieve purity, but this process is laborious, not scalable, and increases the risk of VLP aggregation [20]. Chromatography methods are thus utilized to circumvent these limitations, but these processes may require additional steps in VLP processing, which can also be a disadvantage [20]. Additionally, precautions should be taken to ensure that the formulation of VLP-based vaccines is of high quality [165]. Lastly, conditions such as ion and salt concentrations, as well as pH, must be meticulously controlled during each downstream processing step, as they greatly impact VLP aggregation [20].

After downstream processing, VLPs must be characterized to ensure quality and consistency [20,155]. Before characterization, ammonium sulfate or PEG is employed to precipitate high-molecular-weight VLPs and therefore eliminate low-molecular-weight impurities [20]. It is critical to ensure precipitate purity before moving on to downstream analysis since precipitation can prompt VLP disassembly [20,160]. Biochemical characterization techniques to detect the structural proteins of VLPs include enzyme-linked immunosorbent assay (ELISA), sodium dodecyl sulfate (SDS)-polyacrylamide gel electrophoresis (PAGE), and Western blot (WB) [20,160]. Notwithstanding, through these techniques, it is not possible to distinguish between integral, partially assembled, or unassembled VLPs [20]. Furthermore, these methods display low sensitivity, demand high volumes, and are time-consuming, which impacts downstream studies [160]. Techniques such as high-performance liquid chromatography (HPLC), SDS-capillary gel electrophoresis, and matrix-assisted laser desorption/ionization time of flight mass spectrometry (MALDI-TOF MS) have emerged as alternatives to the classical biochemical characterization methods. Regarding biophysical characterizations, several methods can be explored: transmission electron microscopy to detect particle size and polydispersity; atomic force microscopy (AFM) to determine VLP morphology; and dynamic light scattering (DLS) to assess particle aggregation, among others [160]. Sample preparation for some of these techniques involves drying, which can introduce artifacts into the samples [160]. Cryo-EM can be used to overcome this handicap, as it employs VLPs in solution, but it is only feasible for sufficiently homogenous samples [160]. Lastly, rigorous measurement of particle size and quantification is necessary for correct VLP dosing, which is highly demanding [160]. Approaches such as nanoparticle track analysis (NTA) or electrospray-differential mobility analysis (ES-DMA) can be used to accurately keep track of batch-to-batch consistency, which is particularly useful for VLP-based vaccine formulations [160].

### 4.3. Functionalization

As previously stated, the ability to functionalize VLPs on their inner and/or outer surface is one of the major selling points of these nanostructures [19]. The selected functionalization method can also influence VLP production and function. For instance, genetic fusion is simpler than chemical functionalization and can broaden the applications of VLPs, but it is also associated with low protein yields [166] or inappropriate folding [167]. Chemical conjugation does not have such limitations, but it is more expensive and laborious and typically leads to a stronger immune response, which can be a disadvantage in drug delivery and imaging [168]. It is also important to consider the number of amino acids that are introduced, as well as hydrophobicity, given that these factors are pivotal for adequate VLP structure and function [20,169].

### 4.4. Immunogenicity

One of the main features of VLPs is their capacity to mimic the virus from which they derive and therefore present an icosahedral structure bearing multiple epitopes that activate antigen-presenting cells (APCs) [19,138,161]. VLPs bind to pattern recognition receptors (PRRs) present on the surface of dendritic cells and are consequently taken up for processing [19,138,161]. VLP-derived peptides are then presented on the surface by MHC class I and class II molecules, together with co-stimulatory molecules, for presentation to cytotoxic T and T helper cells, respectively [19,138,161]. Cytotoxic T cells are responsible for the cellular immune response, whereas T helper cells activate both T and B cells, and the latter in turn produce antibodies and are responsible for the humoral immune response [19,138,161]. Interestingly, VLPs are capable of binding directly to B cells and thus provoke humoral responses without requiring the intervention of APCs [19,138,161]. All in all, VLPs are excellent substrates to elicit both cellular and humoral immune responses [19,138,161]. This is an ideal scenario when the end goal is to develop VLP-based vaccines. This becomes a hurdle when VLPs are explored as nanocarriers of imaging probes and/or therapeutic drugs. In vitro studies in cultured cells do not account for immunogenicity, as it does not affect VLP function, but this is a severe hindrance when we consider an in vivo scenario. Tests to assess the immunogenicity of VLPs, together with strategies to reduce it, are thus required and should be explored by researchers. Opting for VLPs derived from viruses with low antigenicity and employing engineering techniques to modify VLPs or replace their surface epitopes are among promising approaches to reduce VLP immunogenicity [170].

## 5. Emerging Approaches and Future Perspectives

VLP applications conjugate different biological fields, namely immunology, virology, microbiology, and vaccinology. Scientists have been drawn to the use of VLPs as special tools to develop novel strategies and methodologies due to their qualities. Genome editing technology, utilizing designer endonucleases, allows for the modification of an organism’s genome or the removal of specific DNA fragments [171]. Early gene editing methods, such as homologous recombination, were inefficient and had high off-target effects [172]. The introduction of designer endonucleases revolutionized this field. The three primary types of designer endonucleases used in DNA modification are CRISPR/Cas, TALEN, and zinc finger nucleases (ZNFs) [173].

VLPs have thus also been explored as delivery vehicles of genome-editing enzymes [174,175,176,177,178]. Long-term expression of genome-editing enzymes has been associated with cytotoxicity, which prompted authors to develop VLPs that can transiently deliver the CRISPR/Cas9 system. Knopp et al. encapsulated CRISPR/Cas in “all-in-one” non-integrating particles of the murine leukemia virus through transfection of *SpCas9* and single guide RNA (sgRNA)-encoding plasmids [177]. Then, transient delivery of these RNAs was performed in multiple murine and human cell lines [177]. Nevertheless, the authors observed that knockout efficiencies were lower than those observed for constitutively active integrating viral vectors [177]. Notwithstanding, stable *SpCas9* expression led to cell arrest and reduced cell metabolic activity and growth, which highlights the advantages of transient delivery [177]. Other investigations have also focused on the transient delivery of CRISPR/Cas technology in VLPs and have shown promising results [174,175,176]. VLP-mediated delivery of other genome editing enzymes has also been described. Cai et al. described the use of HIV-1 Gag particles for the delivery of ZNFs or TALENs by co-packaging the designer endonucleases and donor RNA in “all-in-one” particles. This translated into homology-directed repair and reduced off-target effects [178]. It is important to note that these studies did not report VLP functionalization to achieve delivery, which is something that should be tested to determine whether it is possible to perform tailored genome editing that only affects specific cells by displaying targeting moieties on the VLP surface.

Another emerging approach in the field of nanomedicine is artificial intelligence (AI), which has risen as a potent tool to predict and analyze interactions between different molecules, predict the optimal dosage, drug combination, and administration window for each patient, and ultimately aid in the design and implementation of novel nanomedicines [179]. Li et al. used computer assistance to create a few novel enzymes and achieve catalytic reactions not seen in nature [180]. After redesigning the Bacillus aspartic acid enzyme, several synthetic product-amino acid synthetases that were stereoselective and had perfect position selectivity were produced [180]. Cunningham et al. introduced a customized machine-learning technique, Hierarchical statistical mechanical modelling (HSM) [181]. HSM can reliably predict the affinities of PBD–peptide interactions across a variety of protein families [181]. BANDIT is a Bayesian machine-learning technique for molecule-target predictions, created by Madhukar et al., which can predict drug binding targets [182]. Ianevski et al. have applied DECREASE, a machine-learning model, to assess the effects of combining different medicines while minimizing the number of dose-response measurements, and this has the potential to considerably lower the cost of medication combination screening [183]. Gainza et al. established a conceptual framework based on a geometric deep-learning algorithm, MaSIF (molecular surface interaction fingerprinting), to capture fingerprints relevant to biomolecular interactions [184]. This method allowed the prediction of protein–molecule interactions using chemical and geometric signatures on the surface of molecules [184]. These methodologies are also already very effective tools in the development of VLPs, such as the chimeric VLP based on the Hepatitis B virus core antigen (HBcAg) [185]. With the rapid growth of artificial intelligence, people can use computers to design the structure of VLPs from scratch, which provides a more powerful technical means of enriching the structure of VLPs and expanding their applications [186].

Even with AI’s help, the prediction and design of biomolecular interactions remain difficult despite the recent advancements. For instance, gathering pre-existing data and building a database thereafter takes time. Due to the vast number of proteins and their diverse structures, compiling current information and building a database using a straightforward approach is difficult. After this, more work and verification are still required to use AI to obtain a generalized design scheme based on the data that is already available. Furthermore, it is currently difficult to consider the impact of these variables in the AI-assisted design process since a variety of factors, including the microenvironment and the architecture of the proteins themselves, influence how well proteins operate.

## 6. Conclusions

Nanotechnology, and in particular nanomedicine, has emerged in recent years as a field aimed at addressing the gaps that still exist in traditional medicine, namely in the treatment of diseases such as cancer [187]. Conventional treatments are typically associated with off-target effects and limited delivery to the desired target [187]. Targeted therapy has emerged as a means to tackle these limitations, given that it provides specific delivery to the desired site, minimizing side effects; allows for the delivery of higher doses; and can reach tissues that are typically difficult to access [188]. Among the different delivery vehicles, nanoparticles are of particular interest due to their ability to improve drug solubility and stability, as well as effectively package and deliver different cargoes [3]. Notwithstanding, only a few NP-based formulations (inorganic, liposomal, and polymeric particles) have been commercialized [15,16]. These NPs typically rely on passive targeting and undergo clearance by dendritic cells and phagocytes [15]. Surface modification with PEG has been explored to extend the circulation time of NPs and diminish their uptake by the reticuloendothelial system [15]. The problem is that PEGylation can also increase immunogenicity and reduce target uptake due to the recognition of PEG by the immune system [189]. Within the different types of nanoplatforms, VLPs are a particularly promising alternative to conventional NPs due to their capacity for genetic and chemical modification on their exterior and interior surfaces via amino acid chemistries and self-assembly procedures, which offer precise targeting, solubility, and stability. They are regarded as potent platforms to present antigens, epitopes, polymers, and ligands in a dense repeating array to enhance therapeutic effectiveness. They are also regarded as fitting packaging and delivery platforms for anticancer drugs and genetic materials. For this reason, VLPs have been modified on their surface to achieve targeting and subsequently explored as drug delivery vehicles of moieties such as miRNA [75,108,109,110,111,112] and chemotherapeutic agents [23,75,113,114]. Functionalized VLPs stand out for their increased specificity as nanocarriers for drug delivery. Furthermore, the applicability of VLPs in vaccine formulations has shown that they are considered safe and capable of eliciting a strong immune response against infectious and non-infectious pathologies, namely viral diseases [123,124,125,126,127,128,129,130,131,132,133,134,137] and cancer [44,135,136]. VLPs are also able to bear imaging agents and have been explored as diagnostic agents for optical imaging [140,141,147], MRI [147], and PET imaging [145]. More recently, VLPs have also been explored in biosensing for disease marker detection [153]. Technologies such as genome editing and artificial intelligence could also be explored in the design and production of novel VLP-based formulations. Brought together, these data indicate that VLPs are highly versatile platforms for a wide variety of biomedical applications. The above points show that the field has matured from its early stages; however, there are still obstacles in the way of developing the area of viral medication delivery. The most evident obstacle is whether the viral carrier would be immune to pre-existing or de novo acquired adaptive immunity, and if this would result in side effects or lower efficacy following repeated delivery [58]. Nonetheless, several aspects need to be considered when designing targeting VLPs, especially their immunogenicity, stability, and ability to efficiently reach their target in vivo. Immunogenicity is particularly troubling for drug delivery and imaging, but surface passivation and immune modification techniques have shown promise in lowering the immune response elicited by VLPs [190]. Furthermore, hurdles such as high production costs and scalability hamper the implementation of VLPs in the biomedical setting [160]. Despite these limitations, the potential exhibited by functionalized VLPs far outweighs the presented hindrances, which deems further studies, particularly focused on scalable production and immunogenicity, necessary in the hopes of moving towards in vivo tests and, evenutally, an increasing number of marketed VLPs.

## Figures and Tables

**Figure 1 ijms-25-06699-f001:**
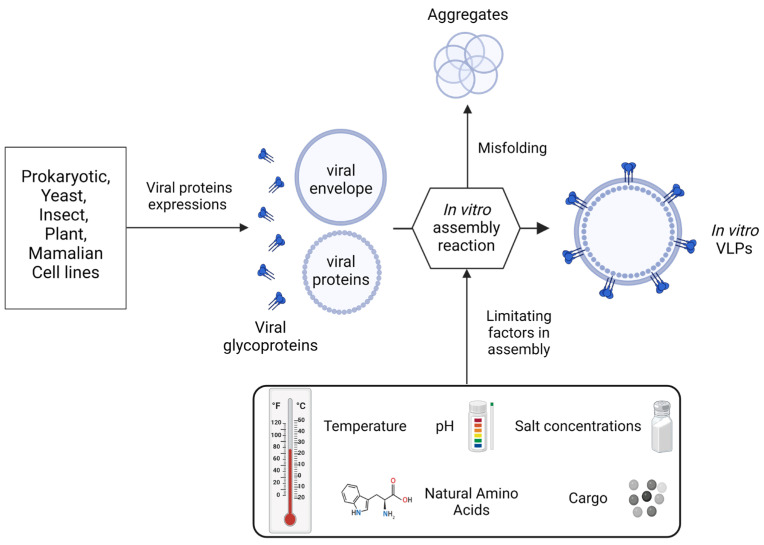
Limiting factors in assembly during the production and design of VLP structures [42,61].

**Figure 2 ijms-25-06699-f002:**
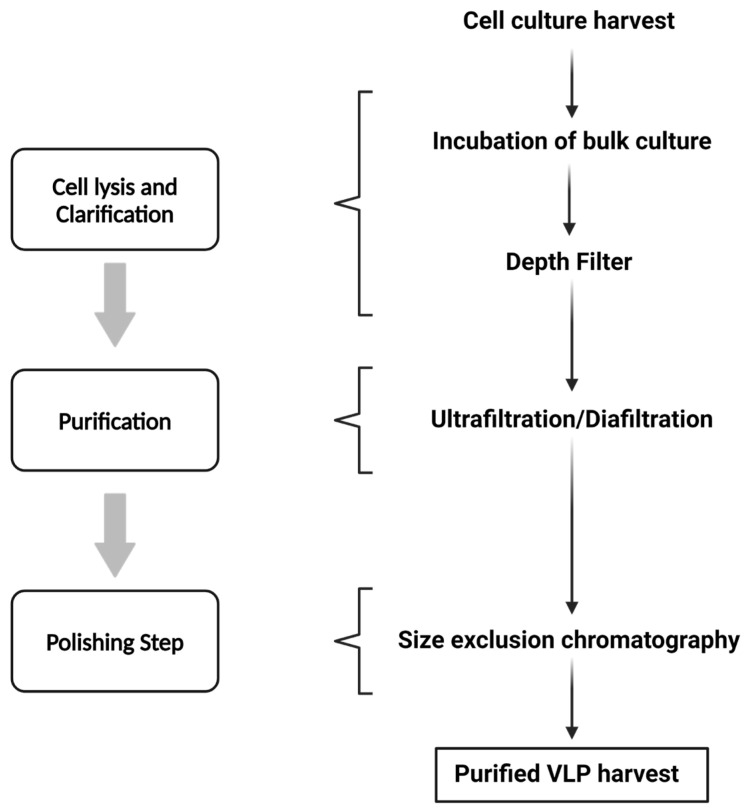
Downstream process approach for VLP purification [73].

**Figure 3 ijms-25-06699-f003:**
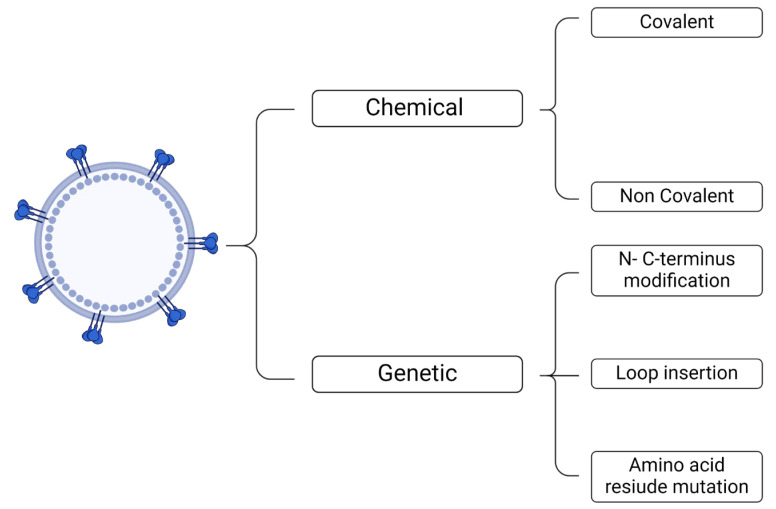
Methods for functionalizing the surface of VLPs.

**Table 1 ijms-25-06699-t001:** Representative table of different enveloped and non-enveloped VLPs, along with their characteristics and applications.

Class	VLP	Virus Type	Virus Protein	Expression Systems	Applications	References
Enveloped	HIV	Human virus	Gag, Env glycoprotein gp120, transmembrane glycoprotein gp41, and polyprotein precursor gp160	Eukaryotic (Mammalian, Yeast, Insect, Plant)	Vaccination and Drug Delivery	[31]
	Influenza	Human virus	polymerase acidic protein (PA), polymerase basic protein 1 (PB1) and PB2	Eukaryotic (Mammalian, Insect, Plant)	Vaccination	[51]
	Ebola	Human virus	nucleoprotein (NP), polymerase cofactor (VP35), (VP40), GP, transcription activator (VP30), VP24, and RNA polymerase (L)	Eukaryotic (Mammalian, Insect)	Vaccination	[52]
	Dengue	Human virus	Capsid (C), envelope (E), and membrane (prM) proteins	Eukaryotic (Mammalian, Yeast, Insect, Plant)	Vaccination and Diagnosis	[53]
	Hepatitis B (surface)	Human virus	Hepatitis B surface antigen (HBsAg)	Eukaryotic (Mammalian and Yeast)	Vaccination	[54]
Non-Enveloped	HPV	Human virus	Protein late (L) 1 and protein late (L) 2	Prokaryotic and Eukaryotic (Mammalian, Yeast, Insect, Plant)	Vaccination and Drug Delivery	[30]
	Norwalk	Human virus	Viral protein (VP) 1 and VP2	Prokaryotic and Eukaryotic (Mammalian, Insect, Plant)	Vaccination	[32]
	Hepatitis B (core)	Human virus	Hepatitis B core antigen (HBcAg)	Prokaryotic and Eukaryotic (Mammalian, Insect, Plant)	Vaccination, Drug Delivery	[55]
	MS2	Bacteriophage	-	Prokaryotic and Eukaryotic (Yeast)	Vaccination, Drug Delivery and Imaging	[56]
	P22	Bacteriophage	-	Prokaryotic	Vaccination, Drug Delivery and Imaging	[57]
	Qβ	Bacteriophage	-	Prokaryotic and Eukaryotic (Yeast)	Vaccination, Drug Delivery and Imaging	[58]

**Table 2 ijms-25-06699-t002:** Representative table of Food and Drug Administration-approved virus-like particles.

Virus	Recombinant Protein	Expression System	VLP Type	Vaccine	Reference
*Hepatitis B virus (HBV)*	HBsAg	Eukaryotic Cell (*S. cerevisiae*)	Non-Enveloped	Engerix-B^®^	[123]
		Eukaryotic Cell *(P. pastoris*)	Non-Enveloped	Enivac HB^®^	[124]
		Eukaryotic Cell (*S. cerevisiae*)	Non-Enveloped	Euvax^®^	[125]
		Eukaryotic Cell (*H. polymorpha*)	Non-Enveloped	Gene Vac-B^®^	[126]
		Eukaryotic Cell (*P. pastoris*)	Non-Enveloped	Heberiovac HB	[127]
		Eukaryotic Cell *(H. polymorpha*)	Non-Enveloped	Hepavax-Gene^®^	[128]
		Eukaryotic Cell (*S. cerevisiae*)	Non-Enveloped	Recombivax HB^®^	[129]
		Eukaryotic Cell (*P. pastoris*)	Non-Enveloped	Revac-B^®^	[130]
		Eukaryotic Cell (*P. pastoris*)	Non-Enveloped	Shanvac-B^®^	[131]
*Papillomavirus*	HPV6/11/16/18 L1	Eukaryotic Cell (*S. cerevisiae*)	Non-Enveloped	Gardasil^®^	[132]
	HPV6 16/18 L1	Insect Cells	Non-Enveloped	Cervarix^®^	[133]
*Influenza virus A*	A (H1N1), A (H3N2), B, HA, NA	Cell-Free	Enveloped	Inflexal^®^ V	[134]

**Table 3 ijms-25-06699-t003:** Different expression systems for VLP production and their advantages and disadvantages [54,160].

Expression System	Advantages	Disadvantages
Bacteria	- Rapid growth - High yields - Easy to manipulate - Low cost - Easily scalable	- Inability to provide PTMs - Cell-derived contaminants
Yeast	- Moderately rapid growth - High yields - Easy to manipulate - Low cost - Easily scalable - Ability to provide PTMs - Low contamination risk	- PTMs with high mannose content (which is immunogenic and decreases the half-life of recombinant proteins in mammalian serum) - Possibility of incorrect assembly and folding
Insects	- Moderately rapid growth - High yields - Easily scalable - Ability to provide PTMs - Ability to produce non-enveloped and enveloped VLPs	- PTMs with high mannose content - Baculovirus/cell-derived contaminants - High cost - Difficult downstream processing
Mammalians	- Ability to provide the most adequate PTMs - Ability to provide authentic VLP assembly - Ability to produce non-enveloped and enveloped VLPs	- Slow growth - Low yields - High cost - Difficult scale-up - Mammalian pathogen contaminants
Plants	- Rapid growth - Low cost - Easily scalable	- Inability to provide PTMs and authentic VLP assembly - Low yields - Low stability (antigen degradation)
Cell-free systems	- High yield - Reduced contaminants - Production of VLPs bearing unnatural amino acids	- Very high cost - Difficult to scale up

## Data Availability

No new data were created or analyzed in this study. Data sharing is not applicable to this article.

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
