# Peer review of "Tailored Viral-like Particles as Drivers of Medical Breakthroughs"

_ijms, 2024, doi:10.3390/ijms25126699_

Round 1

Reviewer 1 Report (New Reviewer)

Comments and Suggestions for Authors

Tailored Viral-Like Particles as Drivers of Medical Breakthroughs

ijms-2982703

 In this review, the authors have addressed the different types of functionalization of VLPs, their importance, and their consequent biomedical applications. VLPs have emerged as promising alternatives to most explored nanoparticles because of the absence of viral genetic material, their incapacity to replicate, mimicry of viral structure, and tropism conservation. VLPs can be surface functionalized with small molecules to improve circulation half-life and target specificity. Through the functionalization and coating of VLPs, it is possible to optimize the response properties to a given stimulus, such as heat, pH, an alternating magnetic field, or even enzymes. 

The article is well written and has thoroughly described various interesting topics within VLP's. The overall flow of the article is perfect as well. The article is perfect and can be accepted in its present form.

Minor comments:

1. Please have a detailed look at the editing of the article.

2. Include more pictures in the article.  

Author Response

Reviewer #1:

Minor comments:

  1. Please have a detailed look at the editing of the article.
  2. Include more pictures in the article. 

Our comment: We have taken the comments into consideration and extensively revised the article. Regarding the inclusion of more figures, we have kept them since the tables (Tables 1-3) are also very illustrative of all the work that has been carried out in the field of functionalized VLPs.

Reviewer 2 Report (New Reviewer)

Comments and Suggestions for Authors

Virus-like particles (VLPs) are promising and versatile tools for therapeutic development, but few designs have been approved. Travassos et al. discuss current and cutting-edge applications of VLPs, including as vehicles for vaccines, biosensing, and drug delivery. The review also explores the challenges of VLP development, especially with expression and downstream processing.

Paper strengths:

Overall, the manuscript is well-written and covers a wide range of research involving VLPs. The authors also discuss everything from proof-of-concept studies to the feasibility of scaling up and implementing VLP production, which helps readers to understand the challenges facing VLPs at multiple stages before approval.

Areas of improvement:

There have been several VLP reviews in recent years, but seemingly none with the breadth of topics present in this manuscript. However, the review as written seems to emphasize breadth over the needed depth for introduced topics. Some paragraphs give extensive background on a topic, then do not spend enough time discussing specific examples on VLPs. The review promises “highlighting the more recent cutting-edge technologies,” but the section on machine learning does not include an example of VLP applications. Furthermore, some aspects of VLP research are omitted from this review, making it less comprehensive. The authors briefly mention AAVs, but do not discuss the research surrounding their application as vectors for gene therapy. Also, how have VLPs been modified in an attempt to improve stability during genetic fusion or other applications (e.g. PP7 dimer, ADDomer)? These research elements would fit into the scope of tailoring VLPs. Specific comments are listed below.

Figure 1: To me, the beneficial characteristics of VLPs would be better explained in the text instead of presented as a figure. If listing characteristics in the text, the authors could give specific examples of these characteristics and how they have been utilized. This format would also allow for a brief explanation of each characteristic, which would aid the understanding of readers.

Lines 84-85: What are some examples of chimeric VLPs? Have they not been studied as much? It would help to address briefly, as this category of VLPs is absent from the rest of the review.

Table 1: Lists some details about VLPs from different viruses, but more information about the composition of these particles would be helpful. What viral proteins form the particles, and are there different forms? Adding a column on this would provide important context. Also, the text right before the table suggests that chimeric VLPs are a separate category, but no examples are provided in the table.

Lines 174-179: The first sentence states that the size of additions to VLPs should be reduced to prevent interference with assembly, but the next two sentences don’t appear to illustrate this concept even though the next sentence leads with “for example”. It would be helpful to clarify how the examples of EGFR and GFP VLPs relate to accounting for size, especially since the sentence at line 174 references peptides instead of protein domains.

Lines 193-201: The beginning of this paragraph seems to introduce information about enveloped VLPs specifically, but several sentences reference studies that involve non-enveloped viruses as mentioned in Table 1 (HBV, QB, MS2).

Line 205: Post-translation modifications are discussed, but the sentence ends “added during the process”. Could you be more specific about the process of obtaining PTMs, and is the process different for different groups? A little detail would help here, as “the process” is a little vague on its own.

Lines 207-208: “Utterly researched” is a phrase that is a little unclear, and there are no corresponding references for this sentence to demonstrate examples of that research.

Lines 209-219: This information about PTMs is thorough, but does not mention VLPs directly and the types of PTMs mentioned are not always discussed in a VLP context later in the paragraph. The authors may consider reducing this section to a few broad sentences about PTMs, as this amount of information felt like it was distracting from specific examples about VLPs relevant to this review.

Line 247: Would be helpful to define HIV TAT protein the first time is it mentioned.

Lines 301-302: “This study” – do you mean these studies? The last four paragraphs have mentioned several studies involving drug delivery and MS2.

Lines 347-351: Wasn’t this study already mentioned in a previous section? Lines 197-201?

Line 506: Ends with “application of a potential” – could be a bit clearer about the method here for this particular assay

Table 3: More detail in the column about the expression system would be helpful. For instance, under “Bacteria” are you referring to E. coli specifically? What cell lines have been used for each type of expression system?

Line 656: If the authors are including a section about the limitations of functionalization, it might help to give a few examples. Are there instances of genetic fusion leading to lower expression for VLPs that have been reported in the literature? How have these limits been explored, and does it vary by VLP?

Lines 691-700: The authors seem to introduce an example of VLPs as a “special tool” for therapeutics, but then only discuss endonucleases and gene editing in this paragraph. Explicitly stating that VLPs have been explored as a delivery vehicle for gene editing enzymes earlier would more clearly introduce this example, while the explanation of the gene editing concept can be condensed.

Lines 720-752: These two paragraphs discuss machine learning and artificial intelligence, but in general terms seemingly unrelated to VLP applications. This information distracts from the stated topic of the review unless specific VLP examples are included.

Line 799: This sentence in the conclusion mentions “regulatory limitations.” Are the authors referring industry hurdles for approval of VLP therapies? If so, this comment does not match the content of the review, as this aspect of VLP development was not addressed.

Comments on the Quality of English Language

See above for specific comments, mainly on clarifying some sentences.

Author Response

Comments

Our responses

Figure 1: To me, the beneficial characteristics of VLPs would be better explained in the text instead of presented as a figure. If listing characteristics in the text, the authors could give specific examples of these characteristics and how they have been utilized. This format would also allow for a brief explanation of each characteristic, which would aid the understanding of readers. 

We have removed figure 1 and listed the characteristics in the text. Furthermore, we have included studies that highlight each feature.

Lines 84-85: What are some examples of chimeric VLPs? Have they not been studied as much? It would help to address briefly, as this category of VLPs is absent from the rest of the review.

We have provided examples of chimeric VLPs by references 51 and 52. However, due to their nature, their surface functionalization has not been described in the literature.

Table 1: Lists some details about VLPs from different viruses, but more information about the composition of these particles would be helpful. What viral proteins form the particles, and are there different forms? Adding a column on this would provide important context. Also, the text right before the table suggests that chimeric VLPs are a separate category, but no examples are provided in the table.

Considering the suggestion made, a column has been added to table 1 with a description of the viral proteins. Regarding chimeric VLPs, since they are different from the others, examples are given in the text below table 1.

Lines 174-179: The first sentence states that the size of additions to VLPs should be reduced to prevent interference with assembly, but the next two sentences don’t appear to illustrate this concept even though the next sentence leads with “for example”. It would be helpful to clarify how the examples of EGFR and GFP VLPs relate to accounting for size, especially since the sentence at line 174 references peptides instead of protein domains.

Considering the reviewer's comments, we have made a more detailed explanation regarding EGFR-derived VLPs to clarify the importance of peptide size in VLP performance.

Lines 193-201: The beginning of this paragraph seems to introduce information about enveloped VLPs specifically, but several sentences reference studies that involve non-enveloped viruses as mentioned in Table 1 (HBV, QB, MS2).

Hepatitis B surface antigen VLPs are described as lipid envelope VLPs (reference 54), which is why they were included in this paragraph. We have included it in Table 1 for clarification. The sentences regarding QB and MS2 VLPs were misplaced and have been moved to the paragraph that refers to non-enveloped VLPs (lines 204-208).

Line 205: Post-translation modifications are discussed, but the sentence ends “added during the process”. Could you be more specific about the process of obtaining PTMs, and is the process different for different groups? A little detail would help here, as “the process” is a little vague on its own.

In view of the comment, we have completed the sentence by adding the observation that the modification occurs by covalent bonding with one or more amino acids.

Lines 207-208: “Utterly researched” is a phrase that is a little unclear, and there are no corresponding references for this sentence to demonstrate examples of that research.

We have changed the phrase to make it more comprehensible. The research about phosphorylation in cardiovascular diseases are described in references 100 and 101, on alzheimer's disease in reference 100 and on diabetes in reference 102.

Lines 209-219: This information about PTMs is thorough, but does not mention VLPs directly and the types of PTMs mentioned are not always discussed in a VLP context later in the paragraph. The authors may consider reducing this section to a few broad sentences about PTMs, as this amount of information felt like it was distracting from specific examples about VLPs relevant to this review.

Considering this comment and the one above regarding the clarification of PTMs, the text has been amended and summarised.

Line 247: Would be helpful to define HIV TAT protein the first time is it mentioned.

We have defined the acronym as requested.

Lines 301-302: “This study” – do you mean these studies? The last four paragraphs have mentioned several studies involving drug delivery and MS2.

We have corrected the sentence.

Lines 347-351: Wasn’t this study already mentioned in a previous section? Lines 197-201?

We detected the issue and corrected it.

Line 506: Ends with “application of a potential” – could be a bit clearer about the method here for this particular assay

To clarify the application of an electrical potential (voltage) to the system during the coulometric measurement process, we have rephrased the text.

Table 3: More detail in the column about the expression system would be helpful. For instance, under “Bacteria” are you referring to E. coli specifically? What cell lines have been used for each type of expression system?

Although specifying the type of bacteria and/or cell lines can be useful in some contexts, in our opinion it is not necessary in this table, since the aim is to explain the general advantages of the different expression systems. Maintaining a general level of detail ensures consistency across the table. This uniformity helps in comparing the overarching benefits of different systems, such as bacteria, yeast, insect, and mammalian cells, without getting bogged down in specific technicalities. In our opinion, those needing specific details can refer to more specialized literature or experimental sections of research papers.

Line 656: If the authors are including a section about the limitations of functionalization, it might help to give a few examples. Are there instances of genetic fusion leading to lower expression for VLPs that have been reported in the literature? How have these limits been explored, and does it vary by VLP?

Considering the reviewer's comment, we have included recent examples of how genetic fusion leads to less expression of VLPs (reference 166) or inappropriate folding (reference 167), as well as an example of how chemical conjugation affects VLP behavior (reference 168). However, we have not found any references associating these limitations with the type of VLP.

Lines 691-700: The authors seem to introduce an example of VLPs as a “special tool” for therapeutics, but then only discuss endonucleases and gene editing in this paragraph. Explicitly stating that VLPs have been explored as a delivery vehicle for gene editing enzymes earlier would more clearly introduce this example, while the explanation of the gene editing concept can be condensed.

Considering the two comments from the reviewer, we have revised the initial part of the text regarding the discussion of endonucleases and gene editing to simplify it without removing the necessary theoretical introduction to understand its innovation in the application in VLPs.

Lines 720-752: These two paragraphs discuss machine learning and artificial intelligence, but in general terms seemingly unrelated to VLP applications. This information distracts from the stated topic of the review unless specific VLP examples are included.

Considering the reviewer's comment, we have included recent examples about artificial intelligence and machine learning (references 185 and 186) to emphasize the importance of these methodologies on VLPs development.

Line 799: This sentence in the conclusion mentions “regulatory limitations.” Are the authors referring industry hurdles for approval of VLP therapies? If so, this comment does not match the content of the review, as this aspect of VLP development was not addressed.

We absolutely agree with the reviewer's comment and have removed the mention of regulatory limitations as it is beyond the scope of this review.

Round 2

Reviewer 2 Report (New Reviewer)

Comments and Suggestions for Authors

The authors have largely addressed my comments. A few additional comments are below.

-Figure 1 was removed, but the figure numbers after it were not adjusted accordingly (starts with Figure 2, there are two Figure 3s)

-A minor point, but the expression system column in table 3 could use more consistency. Bacteria and yeast do not include cells even though those expression systems contain them, but insect, mammalian, and plant do include the word cells. Either add "cells" after yeast and bacteria or simply refer to the expression system (cell-free is fine as is) with the implication that they are cell-based, but it would help to keep the pattern the same.

Author Response

Comment:

-Figure 1 was removed, but the figure numbers after it were not adjusted accordingly (starts with Figure 2, there are two Figure 3s)

Our response: Thank you for your careful analysis of the document, the figures have been renumbered and corrected.

Comment:

-A minor point, but the expression system column in table 3 could use more consistency. Bacteria and yeast do not include cells even though those expression systems contain them, but insect, mammalian, and plant do include the word cells. Either add "cells" after yeast and bacteria or simply refer to the expression system (cell-free is fine as is) with the implication that they are cell-based, but it would help to keep the pattern the same.

Our response: Considering the comment and agreeing with the harmonization of table 3, we refer only to the expression system, removing ‘cell’ from the insect, mammalian and plant systems.

This manuscript is a resubmission of an earlier submission. The following is a list of the peer review reports and author responses from that submission.

Round 1

Reviewer 1 Report

Comments and Suggestions for Authors

The manuscript titled "Tailored Viral-Like Particles as Drivers of Medical Breakthroughs" by Travassos et al. presents a comprehensive review of the development and application of viral-like particles (VLPs) in medical science. The paper aims to showcase the versatility of VLPs in therapeutic and diagnostic settings, highlighting their potential to revolutionize various aspects of healthcare. It delves into the engineering and functionalization of VLPs, their role in vaccine development, targeted drug delivery, and the latest advancements in the field. The authors discuss the unique advantages of VLPs, including their ability to mimic viruses without being infectious, offering a promising platform for safe and effective medical interventions. While the manuscript tackles an important and timely topic, a major revision is necessary by addressing the following points to ensure it provides a significant and novel contribution to the field.

1. The topic of viral-like particles (VLPs) in medical applications is indeed a promising area of research. However, the manuscript needs to distinctly highlight its unique contributions compared to existing literature. Several reviews on VLPs and their applications in drug delivery, vaccine development, and diagnostic imaging have been previously published. The authors need to clearly identify and articulate the novel insights or perspectives their review brings to the scientific community.

2. While the manuscript covers a range of applications of VLPs, it would benefit from a more exhaustive review of recent advancements and comparative analysis of different types of VLPs. Including more recent studies and a broader spectrum of VLP applications could significantly enhance the manuscript's value.

3. The discussion section could be strengthened by providing a more critical analysis of the challenges and limitations associated with VLPs. This includes aspects such as the scalability of VLP production, immunogenicity concerns, and regulatory hurdles. A comparison with other nanocarrier systems could provide a more balanced view of VLPs' position in the field of nanomedicine.

4. The manuscript would benefit from a dedicated section on emerging trends and future directions in VLP research. This could include novel engineering approaches, cutting-edge applications (e.g., in precision medicine or combination therapies), and the integration of VLPs with other technological advancements (e.g., CRISPR, AI in drug design).

5. Expanding on the technical aspects and methodologies involved in VLP design, functionalization, and application would provide more depth to the manuscript. This could include detailed discussions on the genetic engineering of VLPs, surface modification techniques, and the criteria for selecting VLPs for specific applications.

6. Considering the interdisciplinary nature of VLP research, incorporating insights from related fields such as materials science, immunology, and bioengineering could provide a more comprehensive overview of the challenges and opportunities in VLP-based applications.

Comments on the Quality of English Language

Minor editing of English language required

Author Response

Dear Reviewer #1,

Thank you for handling our review manuscript. We would also like to thank the reviewers for the constructive comments, which were determinant for its improvement. We believe that the present version of the manuscript successfully addresses all their concerns. All changes to the manuscript are marked in red in the marked DOCX file. The answers to the reviewer #1 specific comments are included below.

  1. The topic of viral-like particles (VLPs) in medical applications is indeed a promising area of research. However, the manuscript needs to distinctly highlight its unique contributions compared to existing literature. Several reviews on VLPs and their applications in drug delivery, vaccine development, and diagnostic imaging have been previously published. The authors need to clearly identify and articulate the novel insights or perspectives their review brings to the scientific community.

Our comment: Taking the reviewer comment into consideration, we have now included a statement in the abstract that highlights what differentiates our review from previously published reviews.Briefly, in our review we focused in the different types of surface functionalization of VLPs, highlighting the more recent cutting-edge technologies that have been explored for the design of tailored VLPs; their importance, and their consequent applicability in the medical field.

  1. While the manuscript covers a range of applications of VLPs, it would benefit from a more exhaustive review of recent advancements and comparative analysis of different types of VLPs. Including more recent studies and a broader spectrum of VLP applications could significantly enhance the manuscript's value.

Our comment: We have included more recent studies and other VLP applications that highlight the versatility of VLPs and have also added a new section (section 4) that delves into the challenges of VLP-based approaches and depicts a comparative analysis of the different types of VLPs.

  1. The discussion section could be strengthened by providing a more critical analysis of the challenges and limitations associated with VLPs. This includes aspects such as the scalability of VLP production, immunogenicity concerns, and regulatory hurdles. A comparison with other nanocarrier systems could provide a more balanced view of VLPs' position in the field of nanomedicine.

Our comment: We have changed the Conclusions section to Discussion and Conclusions and have addressed the limitations of VLPs and have compared them to other nanocarriers, to provide a more balanced view of the role of VLPs in nanomedicine as requested by the Referee..

  1. The manuscript would benefit from a dedicated section on emerging trends and future directions in VLP research. This could include novel engineering approaches, cutting-edge applications (e.g., in precision medicine or combination therapies), and the integration of VLPs with other technological advancements (e.g., CRISPR, AI in drug design).

Our comment: After Chapter “3. Biomedical Applications of Functionalized VLPs”, a new section (3.4. Biosensing) was included, which focuses on emerging techniques/approaches in VLPs (CRISPR/Cas9 and AI in drug design) and some examples were provided.

  1. Expanding on the technical aspects and methodologies involved in VLP design, functionalization, and application would provide more depth to the manuscript. This could include detailed discussions on the genetic engineering of VLPs, surface modification techniques, and the criteria for selecting VLPs for specific applications.

Our comment: In the Introduction, after the subchapter 1.2 VLP Structure a new subchapter was written, 1.3 VLPs downstream methodologies. This subchapter describes the different steps that are needed after the production of the VLPs for these to advance to preclinical trials.

  1. Considering the interdisciplinary nature of VLP research, incorporating insights from related fields such as materials science, immunology, and bioengineering could provide a more comprehensive overview of the challenges and opportunities in VLP-based applications.

Our comment: Throughout the analysis in the chapters and sub-chapters, challenges have been added to complement what still needs to be worked on to continue improving/advancing in this field.

 Thanking again the careful revision, we hope that the revised version of the manuscript can now meet the reviewers' expectations.

Yours sincerely,

Rita Melo and João Correia.

Reviewer 2 Report

Comments and Suggestions for Authors

The review paper is discussing Tailored Viral-Like Particles as Drivers of Medical Break-Throughs

The review exhibits significant shortcomings in its knowledge and context.

Some comments:

In the abstract, line 14 the authors mention that VLPs have similar (and tropism conservation) to viruses, please be clearer about this conclusion and provide evidence.

As the review is aimed to discuss VLPs, I do not find the first part of the introduction is necessary. Nanoparticles field is a huge field and listing some irrelevant examples of NPs with diverse characteristics does not help the review. Kindly focus on VLPs and their own characteristics and update Table 1 accordingly. 

Figure 1 is very confusing and sending a mixed message of viruses and VLPs. For example: ability to promote transport across cell membrane! Kindly explain what is meant by this character. VLPs are taken up by different immune cells. Capacity to extend circulation times…… This is a contradicting character as the figure also mentioned that VLPs are transported across the cell membrane. The authors should revise all these characteristics of VLPs and also revise the figure itself which represents a virus and not a VLP (the epitopes are confusing).

Lines 57-58, please explain?

Paragraph 66-78: this is all not clear and not entirely correct. It should be revised. 

Figure 2, the authors summarized the limiting factors in very simple way. Please revise and elaborate on these factors. It is important to deliver your point clearly to the readers. 

Line 93: are the authors talking about viruses or VLPs? Please clarify.

Line 107, is this correct? Can you please provide examples of VLPs based on peptides or nucleic acids?

Line 122, can you explain (within the VLPs?).

Line 158, can the authors elaborate on the complications of PTM.

Line 282, this is not correct.

Author Response

Dear Reviewer #2,

Thank you for handling our review manuscript. We would also like to thank the reviewers for the constructive comments, which were determinant for its improvement. We believe that the present version of the manuscript successfully addresses all their concerns. All changes to the manuscript are marked in red in the marked DOCX file. The answers to the reviewer #2 specific comments are included below.

- In the abstract, line 14 the authors mention that VLPs have similar (and tropism conservation) to viruses, please be clearer about this conclusion and provide evidence.

Our comment: VLPs have a similar viral structure to the viruses from which they derive because they use the same viral capsid proteins to self-assemble and they also present tropism conservation due to these same viral proteins, because they have an innate predisposition to target tissues. To fulfil the reviewer's recommendation, we have rewritten the summary to make it clearer in this respect.

- As the review is aimed to discuss VLPs, I do not find the first part of the introduction is necessary. Nanoparticles field is a huge field and listing some irrelevant examples of NPs with diverse characteristics does not help the review. Kindly focus on VLPs and their own characteristics and update Table 1 accordingly

Our comment: Taking the reviewers´ comments into consideration, we have revised the introduction and have removed the examples of NPs. Table 1 has been updated to include the different types of VLPs and their characteristics and has been moved to subsection 1.2.

- Figure 1 is very confusing and sending a mixed message of viruses and VLPs. For example: ability to promote transport across cell membrane! Kindly explain what is meant by this character. VLPs are taken up by different immune cells. Capacity to extend circulation times…… This is a contradicting character as the figure also mentioned that VLPs are transported across the cell membrane. The authors should revise all these characteristics of VLPs and revise the figure itself which represents a virus and not a VLP (the epitopes are confusing).

Our comment:  A new figure was designed to elicit easier perception of the different features that VLPs present for biomedical applications.

- Lines 57-58, please explain?

Our comment: VLPs can be produced with viral proteins from different viruses, such as mammalian viruses, human viruses, insect viruses or plant viruses.

- Paragraph 66-78: this is all not clear and not entirely correct. It should be revised

Our comment: We have revised the paragraphs 66-78 and have made them clearer for the readers.

- Figure 2, the authors summarized the limiting factors in very simple way. Please revise and elaborate on these factors. It is important to deliver your point clearly to the readers

Our comment: A new figure was designed to elicit easier perception of what happens when the following limiting factors are changed for better or worse conditions depending on the type of virus like particles that are being produce from the viral proteins of the viruses.

- Line 93: are the authors talking about viruses or VLPs? Please clarify.

Our comment: In this section we are referring to VLPs and have clarified the text accordingly.

- Line 107, is this correct? Can you please provide examples of VLPs based on peptides or nucleic acids?

Our comment: We aimed to indicate that VLPs can encapsulate and incorporate in on their surface molecules such as peptides or nucleic acids. This was supported with examples provided by other authors as ref 146.

- Line 122, can you explain (within the VLPs?).

Our comment: What we intended to explain in this section is that depending on the location/position of the conjugated peptides or nucleic acids on the surface of VLPs they could elicit different types of reactivity and they could also have lower or higher accessibility to the desired target.

- Line 158, can the authors elaborate on the complications of PTM.

 Our comment: The relevance of PTMs was highlighted by giving examples (ref 94 and 95) of diseases in which PTMs play a pivotal role.

- Line 282, this is not correct.

Our comment: We have changed line 282 to highlight that VLPs can be used as vaccines against their native viruses due to their virus-like structure.

 Thanking again the careful revision, we hope that the revised version of the manuscript can now meet the reviewers' expectations.

Yours sincerely,

Rita Melo and João Correia.

Round 2

Reviewer 2 Report

Comments and Suggestions for Authors

I remain significantly concerned about the accuracy of certain information presented in this review. Without proper discussion and citation, I am unable to accept such claims.

For example:

In the abstract, lines 14-15: The assertion made by the authors regarding VLP tropism towards specific cells is inaccurate.

In the abstract, lines 20-22: It would have been more compelling had the authors solely addressed the diverse surface functionalization of VLPs. Regrettably, the review lacks focus, discussing numerous topics in a superficial manner, thus offering minimal contribution to the understanding of VLPs.

The immunology part in the review is not impressing and very superficial. The authors should have concentrated on their area of expertise, whether it is drug delivery, vaccine production, AI, VLP production, or biosensors.

The figures fail to impart any noteworthy or pertinent insights to the field of VLPs.